# Daily and Seasonal Variation in Light Exposure among the Old Order Amish

**DOI:** 10.3390/ijerph17124460

**Published:** 2020-06-21

**Authors:** Ellen E. Lee, Ameya Amritwar, L. Elliot Hong, Iqra Mohyuddin, Timothy Brown, Teodor T. Postolache

**Affiliations:** 1Mood and Anxiety Program, Department of Psychiatry, University of Maryland School of Medicine, Baltimore, MD 21201, USA; eel013@health.ucsd.edu (E.E.L.); iqramohyuddin@gmail.com (I.M.); 2Department of Psychiatry, University of California San Diego, La Jolla, CA 92903, USA; 3Sam and Rose Stein Institute for Research on Aging, University of California San Diego, La Jolla, CA 92903, USA; 4Veterans Affairs San Diego Healthcare System, San Diego, CA 92903, USA; 5Co-Occurring/Addiction Unit, Sheppard Pratt Hospital, Ellicott City, MD 21043, USA; aamritwar82@gmail.com; 6Maryland Psychiatric Research Center, Department of Psychiatry, University of Maryland Baltimore, Catonsville, MD 21228, USA; ehong@som.umaryland.edu; 7Centre for Biological Timing, Faculty of Medicine, Biology and Health, University of Manchester, Manchester M13 9PL, UK; timothy.brown@manchester.ac.uk; 8Rocky Mountain Mental Illness Research Education and Clinical Center (MIRECC), Veterans Integrated Service Network (VISN) 19, Aurora, CO 80045, USA; 9Military and Veteran Microbiome: Consortium for Research and Education (MVM-CoRE), Aurora, CO 80045, USA; 10Mental Illness Research, Education and Clinical Center (MIRECC), Veterans Integrated Service Network (VISN) 5, VA Capitol Health Care Network, Baltimore, MD 21201, USA

**Keywords:** seasonal affective disorder, Amish, photoperiod, circadian, diurnal, sleep–wake cycles, actigraphy, melanopic illuminance, photopic illuminance

## Abstract

Exposure to artificial bright light in the late evening and early night, common in modern society, triggers phase delay of circadian rhythms, contributing to delayed sleep phase syndrome and seasonal affective disorder. Studying a unique population like the Old Order Amish (OOA), whose lifestyles resemble pre-industrial societies, may increase understanding of light’s relationship with health. Thirty-three participants (aged 25–74, mean age 53.5; without physical or psychiatric illnesses) from an OOA community in Lancaster, PA, were assessed with wrist-worn actimeters/light loggers for at least 2 consecutive days during winter/spring (15 January–16 April) and spring/summer (14 May–10 September). Daily activity, sleep–wake cycles, and their relationship with light exposure were analyzed. Overall activity levels and light exposure increased with longer photoperiod length. While seasonal variations in the amount and spectral content of light exposure were equivalent to those reported previously for non-Amish groups, the OOA experienced a substantially (~10-fold) higher amplitude of diurnal variation in light exposure (darker nights and brighter days) throughout the year than reported for the general population. This pattern may be contributing to lower rates of SAD, short sleep, delayed sleep phase, eveningness, and metabolic dysregulation, previously reported among the OOA population.

## 1. Introduction

Human physiology and behavior exhibit seasonal changes during the fall–winter time interval, including lower mood, reduced energy, sleepiness, decreased interest in social interactions, and increased preference for energy-rich starchy foods, leading to possible weight gain, with spontaneous resolution in spring or summer [1,2]. In certain individuals, seasonal changes in mood include episodes of depression during winter that can be improved with bright light exposure [1,3,4,5]. Seasonal and daily rhythms govern metabolic processes; therefore, alterations in these processes can have an impact on manifestations of metabolic conditions, such as diabetes, and cardiovascular risk [6,7]. The underlying mechanisms of these health outcomes and of physiological and behavioral seasonal changes could be the consequence of seasonal elongation of the duration of the nocturnal melatonin secretion in response to the shortened photoperiod [8], or delayed circadian rhythms due to progressively reduced exposure to morning light during fall and winter [9,10].

A large body of literature now indicates that the effects of light on physiology and behavior such as those described above (so-called non-image forming (NIF) responses) originate via a specific class of intrinsically photosensitive retinal ganglion cell (ipRGCs) [11,12]. The ipRGCs integrate phototransduction via their own photopigment, melanopsin, with synaptic input from rods and cones, and relay signals to the master circadian clock in the hypothalamic suprachiasmatic nuclei as well as other subcortical brain sites [13,14,15]. This arrangement enables ipRGCs to detect the pronounced changes in ambient illumination that occur across the solar day and coordinate downstream physiological responses accordingly. Mood-modulating effects of light are therefore believed to be mediated by ipRGCs, acting via effects on the biological clock, neuroendocrine function, serotonergic tone, and/or more direct impacts on the arousal state [16,17,18,19,20,21,22].

Given the growing awareness of the NIF system’s wide-ranging influence, there is now significant interest in understanding the extent to which ‘unnatural’ patterns of light exposure, associated with modern lifestyles, negatively impact health and well-being. Currently, there is an abundance of literature linking shift work as well as night-time artificial light exposure with a variety of negative health outcomes, including increased risk of developing metabolic disease [23], various forms of cancer [24,25,26], and mood disorders [27].

A key unresolved question, however, is the extent to which alterations in the daily and seasonal patterns of light exposure, associated with modern life, have led to impaired physical and psychological health in the general population [28]. Hence, increased night-time light and reduced daytime exposures to natural sunlight, resulting from increasingly dominant indoor occupational patterns and generalized access to modern electric lighting, are expected to alter the diurnal variations in ipRGC output. This output is required to appropriately coordinate daily physiological rhythms [29]. Similarly, increased evening exposure to short-wavelength light from LED lights, televisions, tablets, tablets, and smartphones [30] is believed to be particularly disruptive to circadian function [31,32]. Indeed, studies in community settings have reported how night-time artificial light exposure is associated with obesity, elevated blood pressure, and depression [33,34,35].

The Old Order Amish (OOA) are a unique population that can help assess the relationship between modern patterns of light exposure and health. The OOA are a predominantly agrarian society using non-grid-fed lights, and do not use artificial time cues with watches or alarm clocks [36]. Thus, their daily patterns of light exposure are expected to differ qualitatively and quantitatively from the general population. The OOA have lower incidence of seasonal affective disorder (depressive disorder triggered by seasonal changes) (<1%) [37,38], lower rates of diabetes [39], and different seasonal-adjusted sleep patterns (with lesser proportion of short sleepers, and earlier wake and sleep onset time) [40] relative to non-Amish. It is tempting to speculate that such differences between the OOA versus the general North American population may be mediated, at least in part, by differences in daily and/or seasonal patterns of light exposure between them. 

Comparisons of quantitative and qualitative ambulatory light exposure, as well as comparisons of activity patterns in OOA with modern counterparts, may provide clues to understanding the low prevalence of metabolic and seasonal mood problems in the OOA community. Here, we address this question through a preliminary study that measures seasonal variations in daily activity and ambulatory light exposure (intensity, wavelength, and timing) among the OOA, via wrist-worn monitors. This study primarily aimed to assess seasonal changes in daily light exposure among the OOA and their association with physical activity. We hypothesized that physical activity and the amount and spectral content of light exposure would vary according to photoperiod.

## 2. Materials and Methods

We investigated daily patterns of activity and ambulatory light exposure across seasons in subjects from an OOA community in Lancaster, PA (40°02′23″ N, 76°18′16″ W). Lancaster is a rural community which experiences wide seasonal variation in day length (9.15 h on 21 December to 14.81 h on 21 June). In this county, the OOA engage mainly in agricultural activities or traditional businesses, such as woodworking. As their religion prohibits the use of grid-fed electric lighting in their home, the OOA use light sources that are not encountered in more modern homes (work spaces are exempt and can use grid-fed lighting). Most houses have portable kerosene or naphtha gas lamps, which can also be portable to provide sources of illumination for the family outside of daylight hours as needed. In addition, many OOA increasingly use battery powered LED lamps for individual activities, such as knitting or reading (information gathered during home visits and interactions with OOA liaisons at the Amish Research Center, Lancaster).

### 2.1. Study Design

Forty physically and psychiatrically healthy subjects (age range 25–74 years, 65% females, mean age 53.5) provided informed consent to take part in a study of individual ambulatory measurement of light exposure. This study was approved by the Institutional Review Board of the University of Maryland School of Medicine (Project ID code HP-00053107, Approval date 30 July 2012). Subjects wore a wrist-worn actimeter/light logger (Actiwatch Spectrum; Philips Respironics, Bend, OR, USA) for 3 consecutive days during two different periods during the year: winter/spring (15 January–16 April) and spring/summer (14 May–10 August). Participants were instructed to wear the watch on their non-dominant wrist and to avoid covering the Actiwatch Spectrum with clothing. Overall, study adherence was high with 33 subjects providing >2 days of continuous light/activity data for both winter/spring and spring/summer conditions. Data from the remaining 7 subjects were excluded from the present analysis for not completing both phases of this study (*n* = 5) or providing incomplete data for at least one phase (*n* = 2; due to removing the Actiwatch Spectrum or covering the device with clothing).

### 2.2. Light and Activity Measures

The Actiwatch Spectrum device (Phillips Respironics, Bend, OR, USA) combined accelerometer-based activity detection with three independent photodiode-based light sensors, with peak response in blue, green or red regions of the spectrum (full width at half maximal response, respectively: 395–500, 475–550 and 600–695 nm; [41]). We recorded activity (arbitrary counts) and RGB light exposure (μW/cm^2^) in 1 min bins across the tested periods. Photopic light exposure (lux) was estimated by the Actiwatch software as a weighted function of the RGB signal. The Actiwatch software also provided automatic detection of sleep onsets/offsets based on periods of inactivity >15 min. After careful inspection of the data from all subjects, we were able to perceive that the automated scoring was reliable in identifying a main ‘sleep’ bout during the night, with only a few occurrences of significant nocturnal activity or apparent daytime napping.

### 2.3. Data Analysis

Our basic analysis approaches for light exposure and activity data were similar (performed using MATLAB, The Mathworks Inc., Natick, MA, USA). Twenty-four hour daily profiles (1 min intervals relative to Eastern Standard Time) were calculated for each individual and photoperiod condition by averaging all the measured values obtained for that timepoint. Sensors on the Actiwatches indicated how long the devices were worn, and rare measurements where the device was not worn were omitted from the resulting average (18,177 data points from a total of 299,336 measurements; ~6% of the data across the two measurement blocks and 33 subjects). Equivalent approaches were used to calculate activity/light exposure relative to wake time or onset of sleep, by referencing the actimetry-defined start and end points of a main nocturnal sleep bout (see above). Similarly, we also calculated daily light/activity profiles as a function of sun position around dawn and dusk (0.2° bins covering solar zenith angles 26° above and below horizon), with sun position calculated as described previously [42]. 

We also calculated each individual’s daily activity/light exposure totals and average values across the following defined epochs: (1) full 24 h, (2) pre-dawn (measurements obtained before civil sunrise: solar zenith 6° below horizon), (3) dawn (between civil and actual sunrise), (4) day (between actual sunrise and actual sunset), (5) dusk (between actual and civil sunset), (6) post-dusk (after civil sunset), (7) 2 h post-wake onset, and (8) 2 h pre-sleep onset. As the precise timing of our winter/spring vs. spring/summer measurements varied across individual subjects, we compared, for each subject, the difference in these parameters measured under long vs. short photoperiod. This was conducted with respect to the difference in civil photoperiod duration (Pearson’s correlation) to assess for seasonal changes. For assessment of light exposure, values were first log-transformed. For clarity and ease of comparison with previous work, absolute values for light exposure were included in the analysis to reflect estimated lux values (as described above). Similar analyses were performed on raw RGB sensor readings and they provided qualitatively similar results (Appendix A).

To assess for daily/seasonal variation in the spectral composition of ambulatory light exposure, we then calculated the fraction of total detected light energy across each channel of the RGB sensor array and analyzed as described above. We also specifically investigated the significance of the detected spectral changes for melanopsin photoreception, whose peak spectral sensitivity (480 nm) is substantially short-wavelength-shifted relative to the photopic visual system (555 nm), by calculating apparent ‘melanopic lux’ [11,43,44]. This was achieved using a recently described method for converting Actiwatch RGB sensor data [45]. The analysis was based on the daylight illuminant model provided in the above study, as this approach is predicted to best account for the majority of light sources to which the OOA are exposed (i.e., either daylight or artificial sources with relatively flat spectral power distributions). The calculated values were then converted to the new, SI-compliant, metric: melanopic Equivalent Daylight Illuminance (units lux) by multiplying by 0.906 as specified in CIE S 026/E:2018 [46]. This correction ensures that for natural daylight (as formalized by the CIE D65 daylight illuminant) melanopic and photopic illuminance are identical. We then compared the derived melanopic illuminance values with photopic illuminance determined by the Actiwatch software.

## 3. Results

The study sample of 40 participants had a mean age of 53.5 years (age range 25–74 years). Twenty-six (65%) of the participants were females. 

### 3.1. Daily Activity Patterns among the OOA

We first examined daily activity patterns among the OOA. Overall, the average daily activity profiles were broadly similar among the participants, with activity starting early in the morning (before 06:30 AM in all but one individual) and persisting for 15.3–20.3 h, with a small dip around midday (Figure 1A; see also Appendix A for data subdivided for subjects tested at timepoints presenting very large or more modest differences in photoperiod duration). Closer inspection indicated that wake onset time almost always occurred before civil sunrise (Figure 1B; *n* = 33/33 subjects during winter/spring and 27/33 subjects during spring/summer) and sleep always occurred after civil sunset. Under both conditions, we also found that activity was reliably higher immediately following wake-onset than preceding sleep onset (Figure 1C; % increase in 1 h totals = 107 ± 18 and 79 ± 18 for short and long photoperiods, respectively; paired *t*-tests both *p* < 0.0005).

Of particular note, we observed a clear photoperiod-related difference in the timing of wake onset (Figure 1D) but not of sleep onset (not shown; r = −0.32; *p* = 0.08), with wake onset timing occurring approximately 1 h earlier under the longest vs. shortest photoperiods tested. Importantly, within-subject comparisons of actimetry-defined mid-sleep (a surrogate measure approximating circadian timing: ‘chronotype’; [47]) revealed a strong correlation between values observed under short and long photoperiods (r = 0.61, *p* = 0.002). These data instill confidence that the sleep–wake onsets defined under our experimental conditions strongly reflect individual sleep onset timing preferences.

In line with the photoperiod-related difference in wake onset timing (but not sleep onset timing), we also found that overall activity levels across the 24 h day significantly increased with photoperiod duration (Table 1). As expected based on the activity profiles described above, we also observed a redistribution of activity relative to the solar day, with increases in total daytime and decreases in total post-dusk activity, in line with increasing photoperiod duration (Figure 1F). Total pre-dawn activity also reduced, albeit to a lesser extent, as photoperiod lengthened (Table 1). By contrast, average levels (counts/min) across all tested epochs were broadly similar, although we did observe a modest trend towards increased daytime and ‘morning’ (2 h post wake onset) activity under long photoperiods (Table 1).

### 3.2. Seasonal Changes in Light Exposure

Next, we evaluated daily/seasonal variation in photopic light exposure (‘lux’). Under both short and long photoperiods, average daytime light exposure was high (>1000 lux), indicating substantial exposure to natural daylight (Figure 2A; Table 2; see also Appendix A for data subdivided for subjects tested at timepoints presenting very large or more modest differences in photoperiod duration). By contrast, during waking hours pre-civil sunrise or post-civil sunset, light exposure was very low (Figure 2B; on average <10 lux), indicative of the relatively dim overall illumination available in Amish homes. Light exposure fell to well below 1 lux during actimetry-defined sleep (Figure 2C). As a population, the OOA experience a very high amplitude of light–dark cycle (with very bright days and dark nights) regardless of photoperiod.

As one would expect, based on the stark differences in the average amount of light exposure detected in daytime relative to that available outside sunlight hours, we also observed clear seasonal differences. Thus, total 24 h and daytime light exposure increased with increasing photoperiod duration, while pre-dawn and post-dusk light exposure decreased (Figure 2D–F; Table 2). We also found a similar relationship with average light exposure across these epochs (Figure 2B, Table 2), i.e., higher average daytime light exposure and somewhat lower average light exposure outside of the civil sunlight hours. We interpret that these data reflect greater time spent outdoors during the spring/summer (both during day and ‘night’), coupled with greater availability of direct sunlight (i.e., less overall cloud cover; see Figure 3C,D below). Nonetheless, insofar as wake onset times in longer photoperiods were closer to sunrise than in shorter photoperiods (Figure 1B), we observed a positive relationship between total light exposure during the 2 h epoch following wake onset and photoperiod duration (Figure 2C; Table 2). Despite the higher average spring/summer light exposure in the 2 h period immediately preceding sleep onset, a direct correlation with photoperiod duration was not detectable across this epoch (Table 2).

### 3.3. Spectral Composition of Light

We also analyzed the RGB sensor data provided by our light loggers to investigate daily/seasonal changes in the spectral composition of light exposure. Under both winter/spring and spring/summer conditions, the proportion of total light energy detected by green and blue sensors decreased and red sensor readings increased in parallel with solar angle around dawn/dusk (Figure 3A,B). Thus, there was a pronounced day–night difference in the spectral composition of light exposure. We did not, however, detect any photoperiod-related differences to the spectral composition of light exposure during total daylight hours (Pearson’s correlation for fractional RGB, all *p* > 0.05; Appendix A). Therefore, there was no apparent difference in the spectral composition of the artificial light sources used by the OOA between winter/spring and spring/summer, nor was there evidence for differences in the proportion of time exposed to natural vs. artificial sources during the day.

Despite the above, as the proportion of the waking hours spent post-sunset increases as photoperiod shortens (Figure 1), the typical spectral composition of the lighting to which OOA are exposed at the end of each day should become relatively long-wavelength-shifted during the winter months. As such, we detected a significant negative relationship between photoperiod length and fractional red light exposure during the 2 h preceding sleep onset (r = −0.39, *p* = 0.03). We also observed a similar relationship when we restricted our analysis to fractional light exposure detected between actual and civil sunset (dusk; r = −0.40, *p* = 0.02; Figure 3A,B). Altogether, these data are consistent with the idea that, during spring/summer months, a greater proportion of evening light exposure comes from natural sources. Equivalent relationships did not attain statistical significance for light exposure in the 2 h period following wake onset or during dawn (r = −0.25 and −0.1, respectively, both *p* > 0.05), suggesting that seasonal differences in artificial vs. natural light exposure are less marked in the morning.

Given the day–night and seasonal differences in the spectral content of evening light, we aimed to understand the impact of these differences on melanopsin photoreception, an important driver of NIF responses [11]. The distribution of melanopic vs. photopic equivalent daylight illuminance (EDI) exposure from all subjects, within and outside of daylight hours as well as under short and long photoperiods, are presented in Figure 3C,D.

Outside of daylight hours, there was a clear difference in overall melanopic vs. photopic EDI under short and long photoperiods, with apparent brightness, as detected by melanopsin being substantially lower than that for the photopic visual system (mean ± SD: −0.37 ± 0.14 log lux; *p* < 0.001). Thus, estimated melanopsin activation provided by artificial light sources used by the OOA was very low (on average <6 melanopic EDI). However, when comparing the distribution of photopic vs. melanopic light exposure outside of daylight hours, we noted evidence of two distinct classes of light source. While, as indicated above, the majority of light exposure was low intensity (~1–30 photopic lux) and with very low melanopsin activity, we also detected a secondary higher intensity peak (>30 lux), where melanopic and photopic lux values were more similar. Based on direct Actiwatch Spectrum sensor readings collected from OOA light sources (not shown) and typical spectral power distributions from such sources [11], these corresponded, respectively, to naphtha/propane lamps vs. battery powered LED lamps sometimes used for reading.

In contrast with the majority of nocturnal light exposure, apparent melanopic and photopic light exposure was generally more similar during the day (albeit still moderately lower for melanopsin: −0.21 ± 0.20 log lux). However, the distribution of encountered light exposure values during the daytime was remarkably broad, with several distinct peaks (including some periods with little access to natural light). Most notably, there was a clear peak in both melanopic and photopic light exposure at very high light levels (~10,000 lux), presumably indicating exposure to direct sunlight (Figure 3D), which is essentially absent in winter/spring (Figure 3C). However, these peaks did not appear to be associated with any pronounced difference in the ratio of melanopic to photopic light exposure relative to other daytime conditions.

## 4. Discussion

We have provided the first comprehensive description of daily and seasonal patterns of light exposure among the OOA. To our knowledge, only one previous study has undertaken a detailed investigation of seasonal patterns of light exposure in human subjects [48]. Moreover, as far as we are aware, our study was the first to investigate spectral changes in light exposure within a community without access to grid-fed electricity. Alongside a growing body of literature documenting daily patterns of light exposure in other populations [48,49,50,51], our findings help to understand how changes in living and working habits across human history have impacted light exposure and sleep habits. In particular, by providing the first report of daily/seasonal light exposure patterns in an essentially pre-industrial-era population for which epidemiological data and documented genetic lineages were readily accessible [37,52,53,54,55], our data provide an important foundation for future studies geared towards deepening our understanding of the relationship between light exposure and human health.

A striking aspect of our findings was that daily patterns of activity and light exposure in the OOA more closely resemble those reported for hunter–gatherer communities in the southern tropics [51] than those living in temperate-zone climates with access to grid-fed electricity. Hence, in accordance with an earlier study [52], we found that the OOA wake up very early, typically before dawn, with actimetry-defined mid-sleep typically occurring between 1 and 2 AM (~2 h earlier than for equivalent populations with free access to electricity; [56]). Moreover, daytime photopic light exposure among the OOA was very high (~4000 lux in spring/summer and 1500 lux in winter/spring). These values were similar to those reported for hunter–gatherer communities and North Americans camping without access to electric light [50,51], but they were approximately 3–8-fold greater than that reported for subjects living in more urban environments [48,49,57]. Similarly, nocturnal light exposure for the OOA (~10 lux) was one-third of that reported for modern indoor electric lighting [48,57,58], which may result in better sleep quality and health. In summary, the OOA experience a substantially more robust day–night difference in light intensity, analogous to pre-industrial era societies [51], than is the norm in modern society, even for outdoor workers [49].

Our data also allowed us to draw inferences about the primary sources of light and their influence on photoreceptive systems. Based on RGB sensory data, we demonstrated that light provided by artificial sources outside of daylight hours was substantially long-wavelength-shifted relative to that experienced during the day. As the photoreceptive systems contributing to human NIF responses (melanopsin and perhaps also S-cones) are maximally sensitive to shorter wavelengths, this nocturnal shift would be expected to further reduce effective light intensity outside of daylight hours. Indeed, we found effective melanopic excitation was ~2.5-fold lower than the corresponding photopic lux values observed across these epochs. Given reported interdevice variability in Actiwatch RGB sensors [41], and error associated with the algorithm used to infer melanopsin excitation [45], these values should only be considered approximate. Nevertheless, the resulting nocturnal values (<6 melanopic EDI), were at the lower end of those required to modulate melatonin secretion and/or reset the circadian clock [59,60,61], and thus less likely to disrupt timing and quality of sleep and wake. 

It should be noted here that, despite the generally lower nocturnal light exposure experienced by the OOA, our basic finding of a long-wavelength shift was qualitatively similar to other recent studies that have measured ambulatory RGB light exposure in homes with electric lighting [57,58]. Thus, we did not identify any direct evidence that the relative amount of short vs. long-wavelength nocturnal light exposure for the OOA differed substantially from those whose homes are lit by commonly used electric sources. However, the light loggers used for these studies might not fully report all relevant light exposure. In particular, televisions and visual displays (laptops, tablets, etc.) are likely to provide a significant source of ocular short-wavelength light exposure for the non-Amish [31], which may not be reliably detected by these wrist-worn devices. By comparison, as OOA do not use such devices, it was less likely that our measurements substantially underestimated nocturnal short-wavelength light exposure in the OOA. Thus, the overall lower light levels the OOA were exposed to outside of daylight hours were likely sufficient to result in a measurably reduced impact on NIF responses [19,61,62,63].

As expected based on the agrarian lifestyle of the OOA [52], and our own data indicating that substantial proportions of their light exposure is from natural sources, we also observed pronounced seasonal variation in light exposure. Despite the overall higher daytime light intensities experienced by the OOA, the magnitude of this change (~3-fold difference in average photopic exposure during winter/spring vs. spring/summer) was broadly similar to that observed in northern Europeans with free access to grid-fed electric lighting [48]. The decreased long-wavelength and increased short-wavelength light exposure during spring/summer evenings were also similar to previous studies. Finally, despite much earlier overall wake up times among the OOA, seasonal variation of wake onset times was similar to that reported for Europeans at a similar latitude [64]. Thus, while the OOA experienced a higher amplitude of day–night differences in light intensity than is typical for modern society, their relative seasonal variation in light exposure did not exhibit overt differences relative to other modern populations.

Given these data, it is worth considering the extent to which specific features of light exposure among the OOA could explain the apparently low prevalence of SAD in this population [37]. We see little evidence supporting the notion that this reflects differences in the relative seasonal variation in light exposure. By contrast, the key differences here are likely due to the markedly greater degree of daytime light exposure during the winter, lower levels of nocturnal illumination, and/or the overall higher amplitude of day–night difference in light intensity experienced by the OOA. 

The decreased nocturnal light exposure may have significant health implications. For example, light exposure at night, even independent of sleep timing, duration and quality, has been implicated in metabolic (obesity and dyslipidemia) and blood pressure abnormalities [35,65]. Similarly, aberrant light exposure has been implicated in depression-like behaviors in animals [18]. Nocturnal light exposure increases the risk of subsequent depression in human studies [33], even when accounting for the duration and timing of sleep. 

A common component of several current hypotheses regarding the etiology of seasonally-mediated mood and metabolic problems is a reduction in the retinal signal driving NIF responses [66], leading to an impaired circadian alignment relative to the external world [9] and/or a reduction in direct arousal-promoting and mood-enhancing actions of light (reviewed in [67]). In either case, the above features noted for winter/spring light exposure among the OOA would be expected to counteract these deficiencies. Specifically, the reduction in light exposure that triggers mood and metabolic issues may be corrected by high daytime light exposure in the OOA, even in the winter. Additionally, reduced exposure to nocturnal light would be expected to ensure robust alignment of the circadian system to the solar day [68]. Of particular note, the relatively strong day–night signal would undoubtedly be expected to oppose the delayed circadian timing that is relatively common among patients with seasonal affective disorder [9,69]. Indeed, in common with previous investigations [52,54], our data certainly indicate a strong morning preference among the OOA. However, the extent to which an association between evening preference and seasonal changes in mood [54,70,71,72] is a contributing factor to developing seasonally-mediated health problems or simply a consequence of the underlying biological mechanisms remains to be further investigated.

Limitations of this study include the absence of sleep and activity diaries as well as mood ratings in our participants. There is a possibility that exposure to light, as well as rest-activity patterns could have been impacted by the mood states of the participants. These analyses did not account for individual-level characteristics such as health status and BMI, which could also influence chronotype and activity levels. Environmental temperature and circadian rhythm-related biomarkers (i.e., melatonin, cortisol, and body temperature) were not assessed during this study, which could potentially impact the behaviors of participants and elucidate the biological mechanisms driving seasonal changes. Finally, the Actiwatch Spectrum was worn on the wrist, and as such, could not precisely assess light exposure at eye level.

## 5. Conclusions

Future studies should include mood ratings to account for bidirectional influences between mood and light exposure. Biomarkers of circadian rhythm changes (e.g., levels of melatonin and cortisol) may also offer unique insight into the mechanisms underlying these health differences in OOA. Recently, a lower frequency of short sleep and a relatively earlier phase of sleep have been reported in the OOA in comparison with non-Amish [40]. These could be, in part, important consequences of light exposure differences, in particular the higher diurnal variation of light exposure, with brighter days and darker nights, potentially contributing to better affective, metabolic and cardiovascular health in OOA relative to non-Amish [39,40,73]. 

We quantified daily variation in activity and ambulatory light exposure across the seasons within a population without access to grid-fed electricity. Collectively, our data support the notion that light exposure patterns typical of the OOA could be protective, at least in part, against conditions characterized by circadian, sleep, affective, and metabolic dysregulation. The well-documented genetic lineages of the OOA, as well as the availability of genetic data and material, and close physical proximity of comparator populations at identical latitudes, present an unparalleled opportunity for future work on genetic, epigenetic, and environmental chronobiological factors and their interactions, with a long-term aim to better understand and target the dysregulation of biological rhythms predictively associated with metabolic, cardiovascular and mental illness.

## Figures and Tables

**Figure 1 ijerph-17-04460-f001:**
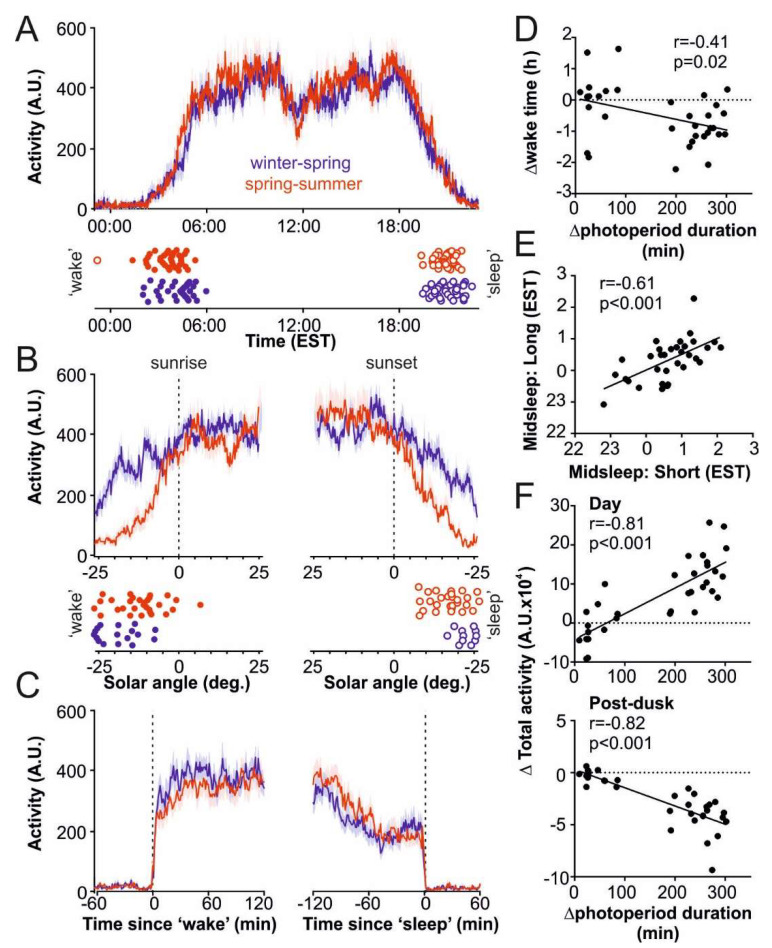
Photoperiod-related differences in activity timing in the Old Order Amish (OOA). (**A**) Mean ± SEM daily activity patterns for OOA subjects during winter/spring or spring/summer; lower panel indicates wake and sleep onset times for each subject (*n* = 33). (**B**) Mean ± SEM activity patterns (upper panels) and wake/sleep onset times (lower panels) for subjects above as a function of sun position around dawn or dusk. (**C**) Mean ± SEM activity patterns for subjects above relative to actimetry-defined wake and sleep onset timing. (**D**) Relationship between seasonal change in wake onset time and photoperiod duration for these 33 OOA subjects. (**E**) Relationship between midsleep timing in these subjects under short and long photoperiods. (**F**) Relationship between seasonal change in total activity during the day (top) or post-dusk (bottom) and photoperiod duration.

**Figure 2 ijerph-17-04460-f002:**
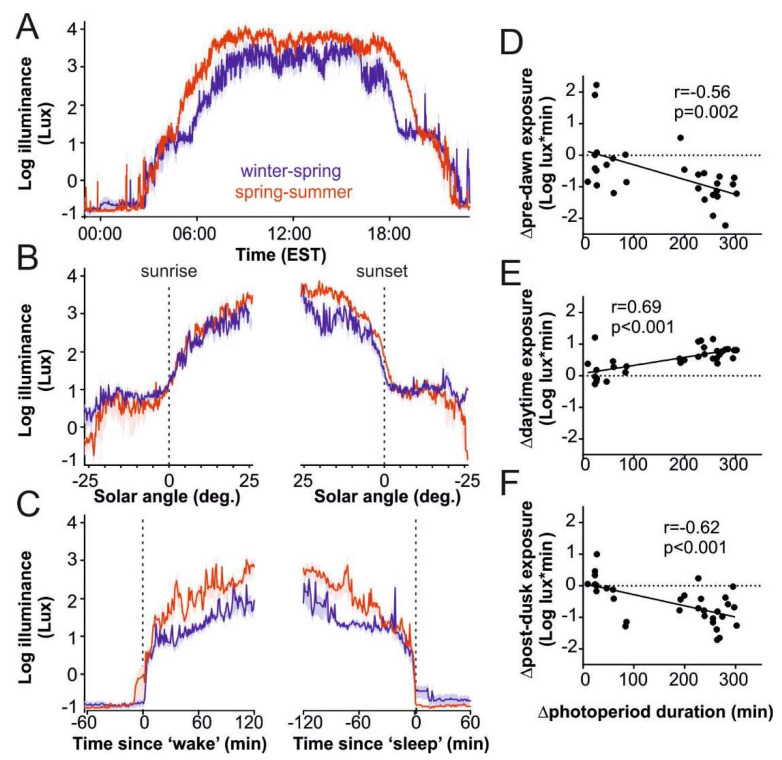
Photoperiod-related differences in photopic light exposure in the Old Order Amish. (**A**) Mean ± SEM daily pattern of photopic light exposure in 33 OOA subjects during winter/spring or spring/summer. (**B**) Mean ± SEM pattern of photopic light exposure for subjects above as a function of sun position around dawn or dusk. (**C**) Mean ± SEM pattern of photopic light exposure for subjects above relative to actimetry-defined wake onset and sleep onset timing. (**D**–**F**) Seasonal changes in pre-dawn (**D**), daytime (**E**) and post-dusk (**F**) total light exposure (lux*min) as a function of difference in photoperiod length for the 33 OOA subjects.

**Figure 3 ijerph-17-04460-f003:**
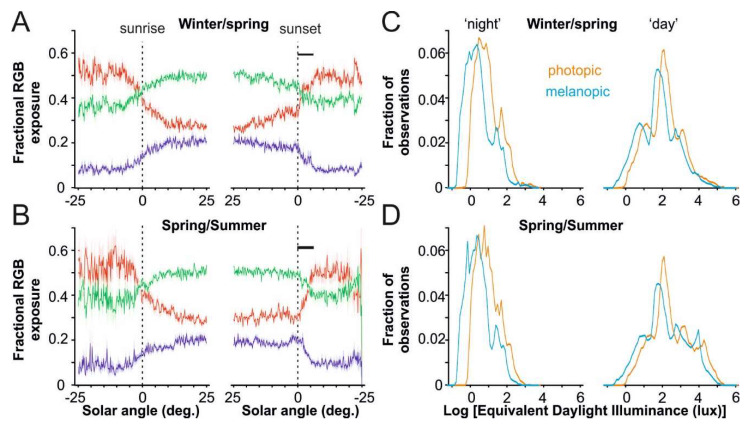
Daily differences in light quality experienced by the Old Order Amish. (**A**,**B**) Mean ± SEM fractional exposure to long (red), medium (green) and short-wavelength light (blue) around dawn and dusk in 33 OOA subjects during winter/spring (**A**) and spring/summer (**B**) months. Photoperiod-related differences were observed only around dusk (actual-civil sunset; indicated by black line). (**C**,**D**) Proportional distribution of apparent melanopic and photopic Equivalent Daylight Illuminance experienced by the 33 OOA subjects outside of daylight hours (‘night’; left) and during the day (right) during winter/spring (**C**) and spring/summer (**D**) months.

**Table 1 ijerph-17-04460-t001:** Activity measures under short and long photoperiods. Shows total and average activity counts (mean ± SD) for each analysis epoch tested under winter/spring (short) and spring/summer (long) photoperiods. Correlation r and *p* values reflect results of Pearson’s correlation for relationship between difference in photoperiod duration (long/short) and activity measures. Analysis epochs are: 24 h total (total daily counts for each individual), pre-dawn (total daily counts from wake onset to civil sunrise), dawn (daily counts from civil to actual sunrise), day (daily counts from actual sunrise to actual sunset), Post-dusk (daily counts from civil sunset to sleep onset), 2 h post-wake onset and 2 h pre-sleep onset (daily counts for first hours post-wake onset and last hours pre-sleep onset). Data for average counts are shown only for those epochs whose duration varies depending on individual and/or photoperiod.

	Total Activity (A.U.)				Average Activity (A.U.)			
	Winter–Spring	Spring–Summer	Correlation with Photoperiod		Winter–Spring	Spring–Summer	Correlation with Photoperiod	
Epoch	Mean ± SD	Mean ± SD	r	*p*	Mean ± SD	Mean ± SD	r	*p*
24 h total	398,944 ± 93,574	411,465 ± 88,517	0.46	0.007				
Pre-dawn	37,192 ± 25,091	15,033 ± 16,434	−0.44	0.011	345 ± 110	221 ± 135	−0.10	0.605
Dawn	10,316 ± 4171	9538 ± 6306	−0.07	0.718	353 ± 140	298 ± 195	−0.14	0.424
Day	284,043 ± 66,078	353,081 ± 89,889	0.81	<0.001	417 ± 99	413 ± 93	0.38	0.03
Dusk	11,869 ± 5587	10,543 ± 5338	−0.02	0.925	407 ± 193	326 ± 157	−0.11	0.531
Post-dusk	41,965 ± 25,289	15,407 ± 10,494	−0.82	<0.001	256 ± 87	187 ± 85	−0.32	0.066
2 h post-wake onset	43,598 ± 12,840	38,358 ± 16,657	0.39	0.03				
2 h pre-sleep onset	28,039 ± 9874	30,621 ± 11,035	0.33	0.06				

**Table 2 ijerph-17-04460-t002:** Light exposure under short and long photoperiods. Shows total (lux values summed over each min of exposure) and average illuminance (mean ± SD) for each analysis epoch tested under winter/spring (short) and spring/summer (long) photoperiods. Conventions as in Table 1.

	Total Exposure (Log lux*min)	Average Exposure (Log lux)
Winter–Spring	Spring–Summer	Correlation withPhotoperiod	Winter–Spring	Spring–Summer	Correlation withPhotoperiod
Epoch	Mean ± SD	Mean ± SD	r	*p*	Mean ± SD	Mean ± SD	r	*p*
24 htotal	6.01 ± 0.29	6.53 ± 0.27	0.72	<0.001	
Pre-dawn	2.93 ± 0.66	2.32 ± 0.87	−0.56	0.002	1.03 ± 0.5	0.63 ± 0.65	−0.44	0.017
Dawn	2.71 ± 0.38	2.28 ± 1.04	−0.18	0.313	1.24 ± 0.38	0.77 ± 1.05	−0.12	0.266
Day	6.02 ± 0.3	6.53 ± 0.26	0.69	<0.001	3.19 ± 0.28	3.60 ± 0.25	0.59	<0.001
Dusk	2.72 ± 0.4	2.97 ± 0.38	0.16	0.372	1.26 ± 0.4	1.47 ± 0.39	0.11	0.531
Post-dusk	3.46 ± 0.48	2.93 ± 0.72	−0.62	<0.001	1.31 ± 0.37	1.15 ± 0.48	−0.37	0.035
2 h post-wake onset	3.44 ± 0.53	3.92 ± 0.73	0.42	0.016	
2 h pre-sleep onset	3.52 ± 0.45	3.87 ± 0.74	0.17	0.356

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
