# Peer review of "Daily and Seasonal Variation in Light Exposure among the Old Order Amish"

_ijerph, 2020, doi:10.3390/ijerph17124460_

Round 1
Reviewer 1 Report
The authors are to be commended for the breadth and depth of this intriguing, innovative, and informative research with a unique population.
Minor suggestions include recognizing May 14th-August 10th as including "Spring" (line 111), providing a clearer description of the results depicted in Table 1, correcting the grammar in line 255 ("this data").
Author Response
We appreciate this feedback and have made these changes. The second time period is renamed “spring/summer” throughout the paper (manuscript and figures) and have updated the Table 1 description for grammar and results.
Reviewer 2 Report
The manuscript presents interesting data similar to that found found for people living in remote parts of South America. While the ideas are not novel the data is useful because it documents the differences in light exposure between people living in cities and people living essentially just off the grid. The paper serves to generate further hypotheses that might be tested in the this population or other off the grid populations, relating to sleep and mood issues.
Author Response
We appreciate the comments from the reviewer and have further noted the hypothesis-generation potential of these findings (Conclusions (Page 16, line 497):
This study presents findings of seasonal light exposure differences among OOA populations that will underlie hypotheses for further research examining mechanisms of sleep and mood issues in this population, as well as other off-the-grid populations.
Reviewer 3 Report
This is a study with the use of wrist-worn monitors in Old Order Amish on daily and seasonal variations in daily activity and light exposure.
The authors clearly describe the limitations of this study. More studies taking into account the secretion of melatonin and/or cortisol in the participants are needed.
The motive of the present study is interesting, and the results may be essential for us to know more about seasonal rhythms in humans. The Old Order Amish participants are a fascinating group for study seasonally, as they are separated from modern society and “ lighting pollution.”
However, I have several concerns in the present study and manuscript.
The details are written as follows.
Major Comments
Introduction
The aim and hypothesis of the study should be more clearly stated.
Methods
I am just wondering about the lack of characteristics of subjects (sex, age, BMI, health status, etc.); the data would be helpful.
All these parameters could influence participants daily activity but also chronotype.
How about environmental temperature? How different was weather between seasons? It could also influence participants behaviour.
Accordingly, the seasonal difference in phase can hardly be explained by exogenous factors?
Did you have thinking about measurement other markers rhythms (melatonin, cortisol, body temperature)? It could improve this study and give additional clear information about daily and seasonal rhythms in OOA.
Minor Comments
Table 1 has duplicate table descriptions
Author Response
Comment 4: Introduction - The aim and hypothesis of the study should be more clearly stated.
Response: We appreciate this feedback and have updated the Introduction (Page 3, Line 102) with the following aim and hypothesis:
“The study aimed to assess seasonal changes in daily light exposure among the OOA and the association with physical activity. We hypothesized that physical activity and the amount and spectral content of light exposure would vary accordingly to photoperiod.”
Comment 5: Methods - I am just wondering about the lack of characteristics of subjects (sex, age, BMI, health status, etc.); the data would be helpful. All these parameters could influence participants daily activity but also chronotype.
Response: We acknowledge this important point and have added the following to the Results section (Page 5, Line 200):
“The study sample of 40 participants had mean age of 53.5 years (age range 25-74 years). Twenty-six (65%) of the participants were females. All participants had no physical or psychiatric illnesses.”
We also noted this in the Limitations (Page 15, Line 483):
“These analyses did not account for individual-level characteristics such as health status and BMI which could also influence chronotype and activity levels.”
Comment 6: How about environmental temperature? How different was weather between seasons? It could also influence participants behaviour.
Response: We agree with the reviewer on the importance of temperature. We did not have data on environmental temperature for this study analyses and have added this to the study limitations in the Discussion section (Page 15, Line 489):
“Environmental temperature and circadian rhythm-related biomarkers (i.e., melatonin, cortisol, body temperature) were not assessed during the study, which could potentially impact the behaviors of participants and elucidate the biological mechanisms driving seasonal changes.”
Comment 7: Accordingly, the seasonal difference in phase can hardly be explained by exogenous factors?
Response: We acknowledge the limitations of the study’s ability to determine all exogenous factors that influence behavior, however the seasonal differences in light exposure have been implicated in altered sleep and activity in non-OOA populations and thus, may provide insight into health differences between OOA and individuals living in grid-fed societies. We have updated the Conclusions section (Page 16, Line 508) to acknowledge these issues:
“Environmental factors may not be the only drivers of mood and cardiometabolic health, so inclusion of personalized risk factors will be important for understanding the links between seasonality and health.”
Comment 8: Did you have thinking about measurement other markers rhythms (melatonin, cortisol, body temperature)? It could improve this study and give additional clear information about daily and seasonal rhythms in OOA.
Response: We have added these biomarker to the study limitations in the Discussion section (Page 15, Line 489):
“Environmental temperature and circadian rhythm-related biomarkers (i.e., melatonin, cortisol, body temperature) were not assessed during the study, which could potentially impact the behaviors of participants and elucidate the biological mechanisms driving seasonal changes.”
We also noted the importance of using biomarkers in future studies to clarify the mechanisms of mood and cardiometabolic health in OOA in the Conclusions section (Page 16, Line 502):
“Biomarkers of circadian rhythm changes (e.g., levels of melatonin and cortisol) may also offer unique insight into the mechanisms underlying these health differences in OOA.”
Comment 9: Minor Comments - Table 1 has duplicate table descriptions
Response: We have updated the Table 1 figure to include make the figure legend more distinct from the manuscript text.
Reviewer 4 Report
In this article, authors studied locomotor activity and lighting conditions of the Old Order Amish (OOA). They compare data collected from winter-spring and summer, and analyzed the effect of seasonal difference of daylength on measured parameters such as activity level, intensity and spectrometric change of light. I felt the concept of the manuscript is interesting but this is “a preliminary study” (p.2 line 89), and has limitations (p.12 line 416) as they admit in the text. As a data collected from unique population of OAA, this paper may attract the interest of the researchers. Therefore, I recommend authors to show all data in their paper (not use “data not shown” but show graphs).
Major
According to Figs 1D, F and 2D-F, delta photo period duration were varied from nearly 0min to 300min. This means data from some participants were collected under almost similar light phase duration between spring-winter to summer, and others were collected at near to the winter solstice and summer solstice. I wonder their analysis is reasonable or not.
From Figs 1D, F and 2D-F, participants seem to be separated in two groups, i.e. small delta photoperiod (0-100min, 12 subjects) and large delta photoperiod (200-300min, 21 subjects). According to graphs shown in Fig.1A-C, and Fig.2A-C, using only large delta photoperiod group for analysis may fit their aims of the study.
Minor
- According to P values, please use <0.001 (less than 0.1%) instead of 2E-06 (or 2E-03, 1E-05…) if journal accepted (Table 1 and 2).
- They use Time (EDT) in Fig.1 and Time (EST) in Fig.2. Why?
Author Response
REVIEWER 4:
Comment 10: In this article, authors studied locomotor activity and lighting conditions of the Old Order Amish (OOA). They compare data collected from winter-spring and summer, and analyzed the effect of seasonal difference of daylength on measured parameters such as activity level, intensity and spectrometric change of light. I felt the concept of the manuscript is interesting but this is “a preliminary study” (p.2 line 89), and has limitations (p.12 line 416) as they admit in the text. As a data collected from unique population of OOA, this paper may attract the interest of the researchers. Therefore, I recommend authors to show all data in their paper (not use “data not shown” but show graphs).
Response: We appreciate this helpful comment from the reviewer and have updated the results to include supplemental Table 1 as well as supplemental Figure 1 and 2.
Comment 11: Major. According to Figs 1D, F and 2D-F, delta photo period duration were varied from nearly 0min to 300min. This means data from some participants were collected under almost similar light phase duration between spring-winter to summer, and others were collected at near to the winter solstice and summer solstice. I wonder their analysis is reasonable or not.
Comment 12: From Figs 1D, F and 2D-F, participants seem to be separated in two groups, i.e. small delta photoperiod (0-100min, 12 subjects) and large delta photoperiod (200-300min, 21 subjects). According to graphs shown in Fig.1A-C, and Fig.2A-C, using only large delta photoperiod group for analysis may fit their aims of the study.
Response: We thank the Reviewer for this suggestion and are happy to include two new supplemental figures (Fig S1 and S2) which present the corresponding analysis from Fig 1A-C and Fig2A-C separated according to groups of subjects that experienced large and more modest differences in photoperiod duration.
Comment 13: Minor. According to P values, please use <0.001 (less than 0.1%) instead of 2E-06 (or 2E-03, 1E-05…) if journal accepted (Table 1 and 2).
Response: We have made those changes in the Tables and Figures as suggested.
Comment 14: They use Time (EDT) in Fig.1 and Time (EST) in Fig.2. Why?
Response: We have updated Figure 1 to use EST as that is more congruent with the OOA.